

# Exclusion process subject to conformational size changes

Yvan Rousset[1], Luca Ciandrini[1,2,3] and Norbert Kern[1*]

**1** Laboratoire Charles Coulomb (L2C), Université de Montpellier, CNRS, Montpellier, France
**2** Centre de Biochimie Structurale, Université de Montpellier,
CNRS, INSERM, Montpellier, France
**3** DIMNP, Université de Montpellier, CNRS, Montpellier, France

★ norbert.kern@umontpellier.fr

## Abstract

We introduce a model for stochastic transport on a one-dimensional substrate with particles assuming different conformations during their stepping cycles. These conformations correspond to different footprints on the substrate: in order to advance, particles are subject to successive contraction and expansion steps with different characteristic rates. We thus extend the paradigmatic exclusion process, provide predictions for all regimes of these rates that are in excellent agreement with simulations, and show that the current-density relation may be affected considerably. Symmetries are discussed, and exploited. We discuss our results in the context of molecular motors, confronting a hand-over-hand and an inchworm stepping mechanism, as well as for ribosomes.

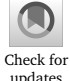
## 1  Introduction

Non-equilibrium statistical physics is a rapidly developing field. In contrast to its equilibrium counterpart built upon the probability distribution of states, no such general framework is available for non-equilibrium systems as the required distributions have not been established yet, even for describing stationary states. Uni-dimensional driven transport models based on particles which actively move on a lattice have been used as prototypical situations, and studying features of such non-equilibrium systems constitutes an attempt to further our knowledge and pose the grounds for a general theory. Besides, these models share many features with traffic systems (biological and other), and they have recently attracted the attention of researchers from many disciplines.

In this work we study a driven lattice gas in one dimension, implemented by an exclusion process featuring particles which advance stochastically via a cycle of conformations characterised by different particle sizes.

The simplest form of an exclusion process is known as TASEP (Totally Asymmetric Simple Exclusion Process). It consists of particles advancing on a discrete lattice, subject only to an excluded volume constraint which limits the occupancy to a single particle on any lattice site. This process has been introduced originally by MacDonald et al. [1,2] to describe the process of translation, where ribosomes move along a strand of messenger RNA (mRNA) to synthesise proteins. It has since been recognised as a minimal model retaining many essential features of out-of-equilibrium transport, which has made it very popular for studying fundamental aspects of stochastic transport [3]. Extensions of the process have been used to describe the dynamics of motor proteins stepping stochastically along biofilaments of the cytoskeleton, such as microtubules or actin filaments [4,5]. Other applications of the model abound, such as for pedestrian dynamics [6] or queuing theory [7,8]. More recently, the initial motivation of translation has regained importance, and models based on exclusion processes have been shown to be useful in analysing complex aspects of the translation process, such as estimating the rate of aborted translation [9], competition for ribosomes from a pool of shared resources [10,11], the role of slow codons [12–14] and many others.

The motivation for our model lies in the fact that there are competing, or complementary, microscopic processes by which a motor protein moves. All types of motors go through a cycle of conformations involving internal states, a fact that has been shown to impact the collective transport process [15–18]. The nature of microscopic conformations associated with each step in the cycle vary significantly between motors [19–21]. Specifically, certain types of motors are thought to advance via 'hand-over-hand' motion, in which parts of the motors (heads) successively step ahead of the other, thereby moving the motor forward. Other types can perform an 'inchworm' motion, in which there is a designated leading head, which steps ahead during the first part of a cycle, then to be followed by the second head.

The point we address here is that the steric occupancy of a motor is therefore bound to vary along the stepping cycle, as the effective excluded volume depends on the microscopic stepping mechanism (see the illustration in Fig. 1). As motion is coupled to crowding, this

impacts the collective transport which can be achieved. We show in the following that the implications are non-trivial, but we elaborate arguments based on mean-field predictions and symmetries which provide a rather complete picture.

In the following we first define an exclusion process subject to the cyclic variation of occupancy along the succession of internal states (Section 2). We then discuss the specific case corresponding to the hand-over-hand motion presented in Fig. 1 in detail, and present a complete picture of the phenomenology, comparing predictions with results from numerical simulations (Section 3). The general model, designed to cover also the case of 'inchworm' motion, is analysed in Section 4. We then discuss our findings, point out their potential impact and explore biological applications related to gene translation (Section 6) before concluding.

## 2 Model

Our model is a driven lattice gas in one dimension implementing an exclusion process in which particles assume two different conformations in the course of their dynamics. The particles can be in a *compressed* ($-$) and in an *expanded* ($+$) state. Compressed particles occupy $\ell_-$ sites of the lattice, while a particle covers $\ell_+ > \ell_-$ sites when found in the expanded state. A sketch of the system is shown in Fig. 2.

A particle in the compressed state can expand with a rate $\gamma_+$ provided that it has space to do so, i.e. there must be at least $\Delta\ell := \ell_+ - \ell_-$ empty sites ahead of it. If this condition is satisfied, the particle advances by expanding. An expanded particle then transitions to the compressed state with rate $\gamma_-$, without the need of satisfying any other condition. It thus frees the $\Delta\ell$ trailing sites previously occupied by the particle.

We define the dimensionless parameter

$$R := \frac{\gamma_+}{\gamma_+ + \gamma_-}, \qquad R \in (0, 1),$$

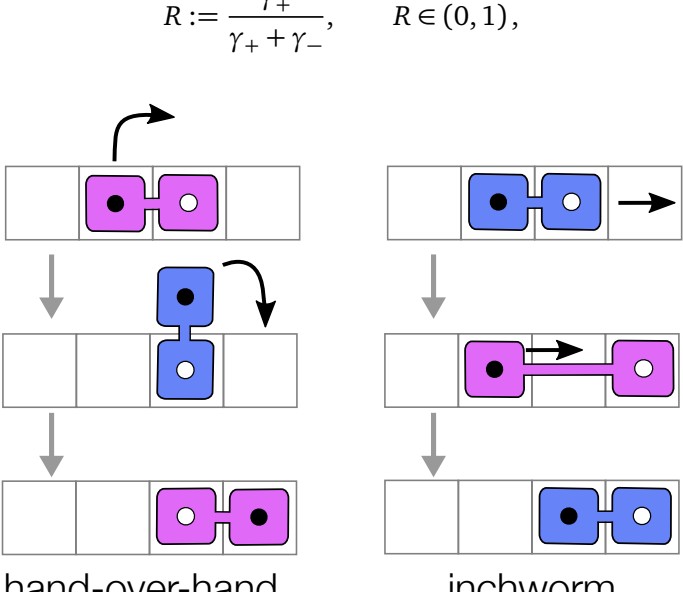

hand-over-hand          inchworm

Figure 1: Illustration of the footprint of molecular motors with two active heads ($\bullet, \circ$), according to the microscopic configurations during their stepping cycle. Starting from the same configuration (top), a 'hand-over-hand' motor (left) cyclically occupies one or two sites, whereas 'inchworm' motion (right) implies that the occupancy oscillates between two and three sites. Different colours distinguish the 'contracted' and 'expanded' configurations in the two dynamics.

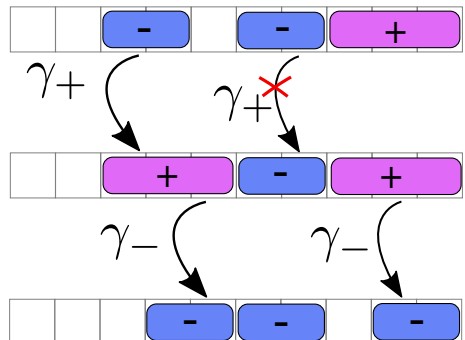

Figure 2: Sketch of the model with $\ell_- = 2$ and $\ell_+ = 3$, showing how configurations can evolve. Movement is from left to right. Expanded particles (+) contract with rate $\gamma_-$, whereas contracted particles (−) expand with rate $\gamma_+$ provided there are at least $\Delta\ell = \ell_+ - \ell_-$ empty sites ahead of them.

which will allow us to distinguish between regimes of *slow expansion* ($\gamma_+ \ll \gamma_-$, i.e. $R \simeq 0$) and *fast expansion* ($\gamma_+ \gg \gamma_-$, i.e. $R \simeq 1$). The intermediate regime, where the timescales of expansion and compression are similar ($\gamma_+ \simeq \gamma_-$), corresponds to $R \simeq 1/2$. We also notice that $R$ directly indicates the probability of finding a single isolated particle in the (+) state, and thus $1-R$ is the complementary probability of finding an isolated particle in the (−) state.

In the next section we will study the collective movement of particles on a periodic lattice with $L$ sites. Due to the periodic boundary condition, the total number of particles in the system is fixed to $N$. The number of particles in the expanded ($N_+$) and compressed ($N_-$) states are determined by $R$, but also by steric effects. We define the *partial particle densities* as

$$\rho_\pm := \frac{N_\pm}{L},\tag{1}$$

such that the total density

$$\rho := \rho_+ + \rho_- = \frac{N}{L}\tag{2}$$

is constant due to particle conservation. We furthermore introduce the density of unoccupied sites, which we note $\rho_|$:

$$\rho_| = 1 - \ell_- \rho_- - \ell_+ \rho_+.\tag{3}$$

Throughout this work we will use these particle densities, in which all particles are accounted for, irrespective of their size. The total density will in general not reach unity as it is bounded by $\rho \leq \ell_-^{-1} \leq 1$. We distinguish these (number) densities from *coverage* densities $\eta_\pm$, which represent the percentage of lattice sites which are effectively occupied by particles in state (+) or (−): $\eta_\pm = \ell_\pm \rho_\pm$ and thus $\eta = \ell_+ \rho_+ + \ell_- \rho_-$. These behave like volume fractions, and are bounded by 1.

The current can in principle be defined based on counting the displacements of a marker anywhere on the particles, but the choice will affect how the expansion and contraction steps contribute to the current ($J_+$ and $J_-$, respectively). Here we will consider the centre of gravity, so that each half-step corresponds to a displacement of $\Delta\ell/2$ sites, and yields identical contributions to the total current:

$$J = 2J_+ = 2J_- .\tag{4}$$

# 3 Hand-over-hand motion ($\ell_- = 1, \ell_+ = 2$)

We first discuss the phenomenology of the model for the case $\ell_- = 1, \ell_+ = 2$, which corresponds to the simplest scenario of hand-over-hand motion. In this example particles cycle through a succession of compressed states (occupying $\ell_- = 1$ sites) and expanded states (with $\ell_+ = 2$ sites). This is the case closest to the standard TASEP, but even here we shall see that new features arise, with the exception of the $R \to 0$ limit. We establish an approach which we then generalise to an arbitrary choice of $(l_-, l_+)$ in Section 4.

## 3.1 Phenomenology of simulations

We first survey the phenomenology based on simulation data - the numerical procedure is described in Appendix B. Figure 3 shows graphs of the current-density relation $J(\rho)$, superposing the different regimes we expect to observe in terms of the expansion/contraction rates: $R \simeq 0$, $R \simeq 1/2$ and $R \simeq 1$.

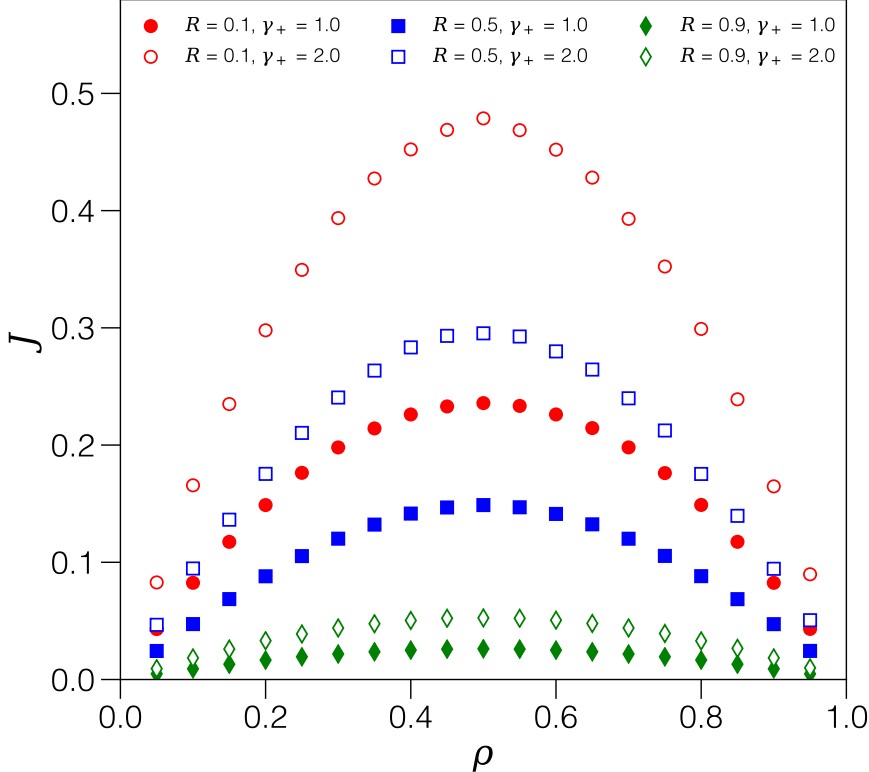

Figure 3: Current $J$ as a function of the particle density $\rho$ for different values of the parameters $R$ and $\gamma_+$.

A first observation is that, for all regimes, there appears to be a particle-hole symmetry. This is a well-known feature of the simple TASEP model, but may come as a surprise here: indeed, our model carries elements both of an internal state [16, 17] and of particle sizes exceeding a single lattice site [1, 22]. Both models have been shown not to obey particle-hole symmetry, yet the combined model restores the symmetry. This is particularly intriguing as both models skew the current-density relation towards higher densities, so this is not a simple compensation of effects. We will discuss the particle-hole symmetry below, and show that it is specific to the choice ($\ell_- = 1, \ell_+ = 2$).

To better characterise the behaviour it is useful to analyse the partial densities $\rho_\pm$ and the hole density $\rho_|$ as a function of the total density $\rho$. In particular the density of expanded

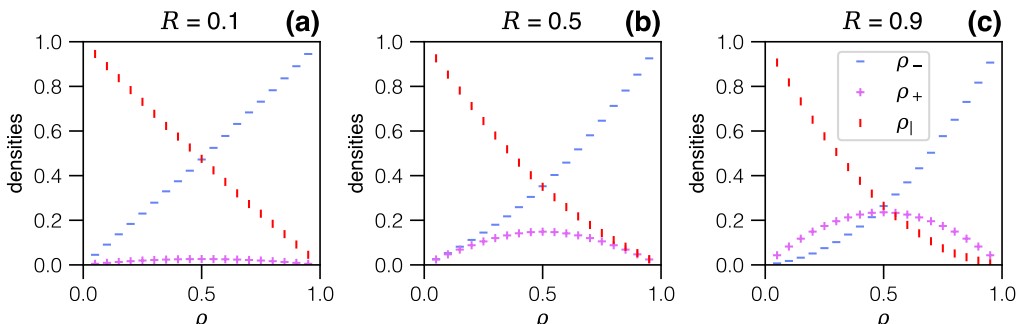

Figure 4: Plots of the partial densities of expanded particles ($\rho_+$), compressed particles ($\rho_-$) and empty sites ($\rho_|$) as a function of the overall particle density $\rho$, for different values of $R$: (a) expansion-limited regime, (b) intermediate regime and (c) fast expansion.

particles $\rho_+$ is directly related to the current: as the compression step is not subject to steric exclusion, the expansion current is directly related to $\rho_+$ as

$$J_- = \frac{\Delta\ell}{2}\gamma_-\rho_+,\tag{5}$$

with $\Delta\ell = 1$ in this case. The total current is then just double this contribution (see Eq. 4).

Therefore knowing $\rho_+$ is equivalent to knowing the current but with the additional advantage, as we will show, that $\rho_+$ depends on the single parameter $R$, whereas the current explicitly depends on both rates $\gamma_+$ and $\gamma_-$ (compare symbols with the same shape in Fig. 3). We are thus particularly interested in $\rho_+$, but the other partial densities also help to understand the process: all densities are plotted in Fig. 4 for all three regimes. We first focus on these to discuss the underlying phenomenology, on the basis of which an analytical description will then be established.

In the expansion-limited regime ($R \simeq 0$) in Fig. 4.a, $\rho_-$ grows (almost) linearly with the overall density, whereas $\rho_+$ remains exceedingly small: due to fast contraction, any particle having undergone expansion will almost immediately re-contract. The particle-hole symmetry is apparent from the simulation data for $\rho_+(\rho)$, the maximum of which thus arises at $\rho = 1/2$, but also in the complementarity of $\rho_-$ and $\rho_|$, as the data suggests that $\rho_-(\rho) = \rho_|(1-\rho)$. Since in this regime particles do not remain in the expanded state for any significant length of time, the current is essentially identical to that of a TASEP model where particles simply hop forward, albeit with a rate which is dominated by the slowest step, which here is the expansion.

The picture changes for the regime where the expansion and contraction rates $\gamma_\pm$ are comparable ($R \simeq 1/2$, in Fig. 4.b). Here the density of expanded particles $\rho_+$ is higher: the expansion rate no longer constitutes a bottleneck, but at the same time expansion is self-limited due to steric hindrance as more expanded particles appear. The current maximum still occurs at $\rho = 1/2$, which is somewhat counter-intuitive as there is additional crowding when compared to TASEP.

In the regime of fast expansion ($R \simeq 1$, in Fig. 4.c) one should expect expanded particles to dominate, but this is again counteracted by steric effects. The hugely more complex phenomenology in this regime is highlighted by the current maximum, still at $\rho = 1/2$: at this point $\rho_+ = \rho_- = \rho_|$, i.e. expanded particles, compressed particles and free sites all arise with comparable probability, and they become equally likely in the limit $R \to 1$. We are thus unavoidably dealing with a mix of particle sizes and, in contrast to the first regime of slow expansion, no simple description seems possible.

### 3.2 Particle-hole symmetry

The data for the current-density relation $J(\rho)$ or, equivalently, for $\rho_+(\rho)$, clearly suggests a symmetry $\rho \leftrightarrow 1-\rho$. In regular TASEP this is due to a particle-hole symmetry, i.e. the fact that a current can likewise be attributed to advancing particles or to receding holes, and particles and holes obey the same dynamical rules. In TASEP this is easy to see, as any move is a simple exchange in position of a particle and the hole ahead. Looking at Fig. 2, no obvious symmetry appears to be present, and indeed in general there is no such symmetry between particles and holes. This may also seem to apply to the case $(\ell_-, \ell_+) = (1, 2)$, represented Fig. 1.a, as an empty site temporarily disappears during the stepping cycle. However, it suffices to consider that the two sites carrying the expanded particle simultaneously carry an expanded hole. When the dynamics is sketched in terms of these objects ( '-' for a compressed particle, '|' for an empty site, and '+' for the combined object of an expanded particle and an expanded hole), as sketched in Fig. 5.b, then it becomes clear that the symmetry is restored in this particular case.

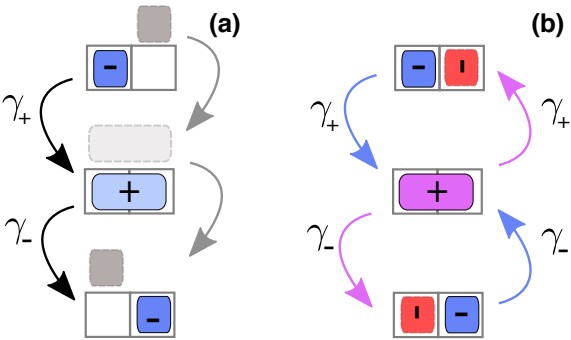

Figure 5: (a) Visualisation of the $\ell_- = 1, \ell_+ = 2$ process, also as seen in terms of holes (in gray). (b) The symmetry becomes apparent when the extension step of a $(-)$ particle is assimilated to the creation of a new object $(+)$, which has characteristics of both an expanded particle and an expanded hole: the process in which holes ($|$) recede obeys exactly the same dynamics, and the same rates, as particles advancing, which implies the particle-hole symmetry. The arrows evolving from bottom to top are exploited below (section 3.5).

    Several remarks are in order. First, it is clear that no such equivalence can be established for a regular TASEP process with extended particles, as considered by Shaw et al. [22], as there is no intermediate phase in the stepping process to which expanded holes could be attributed. Second, the notion also does not apply to a TASEP with an internal states but of fixed size [16, 17]: at first sight it might seem that introducing 'activated holes' should be sufficient, but closer inspection shows that the rules for activating holes are different from those of activating particles (holes only get activated when they directly follow a particle, whereas particles can get activated anywhere), and therefore there is no particle-hole symmetry in this model either. Finally, the symmetry $\rho \leftrightarrow 1 - \rho$ is not present in the general case of particle sizes cycling between $\ell_-$ and $\ell_+$: in any process with $\ell_- > 1$ there no longer are particles of size 1, but holes of size 1 are still required for describing the dynamics, and therefore there can be no such symmetry. The same applies to the complementary case, $\ell_- = 1$ and $\ell_+ > 2$, where one particle of size 1 has a counterpart of several holes, such that it is impossible to swap their roles.

### 3.3 Straightforward mean-field analysis

Straightforward mean-field predictions work extremely well for standard TASEP, but we will see that this is not the case here. To establish a prediction we write the current in terms of the expansion step as

$$J = 2J_+ = \gamma_+ \rho_- \rho_| \,, \tag{6}$$

which is the mean-field expression reflecting the fact that any particle potentially going to expand (probability $\rho_-$) must find an unoccupied site ahead (probability $\rho_|$). Equating the number of successful expansion and contraction events per unit time, and after expressing $\rho_-$ as a function of $\rho_+$ using Eq. (2) for $\ell_- = 1$ and $\ell_+ = 2$, this can be written as

$$\gamma_- \rho_+ = J = \gamma_+ (\rho - \rho_+)(1 - \rho_- - 2\rho_+) \,, \tag{7}$$

which constitutes a quadratic equation for $\rho_+$ as a function of the overall density $\rho$. Picking the negative branch, which is the only physical solution, we have

$$\rho_+ = \frac{J}{\gamma_-} = \frac{1}{2R}\left(1 - \sqrt{1 - 4R^2 \rho (1-\rho)}\right). \tag{8}$$

This mean-field prediction is compared to simulation data in Fig. 3. As expected, it is very accurate for $R \simeq 0$, works less well for $R \simeq 1/2$, but entirely fails for $R \simeq 1$. This corroborates the discussion above, showing that the compression-limited regime is qualitatively different and much more complex.

### 3.4 Improved mean-field analysis

Deviations from this straightforward mean-field expression are expected if correlations are present, and it is typically difficult to formalise these in order to obtain improved predictions. Fortunately, there is a simpler argument for our case, which consists in determining the (average) probability $P_+$ for a compressed particle to be able to perform an expansion, so that the associated current can be written as

$$J_+ = \gamma_+ \frac{1}{2} \rho_- P_+ \,, \tag{9}$$

where the factor $1/2$ stems from the $\Delta\ell/2$ step of the centre of gravity. To evaluate $P_+$ we proceed by mapping the instantaneous lattice configuration onto a reduced TASEP-like lattice in which the expanded particles have been reduced to size 1, thereby eliminating those sites that do not participate in the dynamics. The acceptance probability of an expansion step is then identical to the probability for a particle hop to succeed in regular TASEP dynamics on this reduced lattice. In this reduced system the corresponding reduced density is

$$\tilde{\rho} = \frac{N}{L - N_+} = \frac{\rho}{1 - \rho_+} \,, \tag{10}$$

and thus the current reads

$$J = \gamma_+ \rho_- (1 - \tilde{\rho}). \tag{11}$$

Equating, as before, the contributions from expansion and expression steps, we have

$$\gamma_- \rho_+ = J = \gamma_+ (\rho - \rho_+)\left(1 - \frac{\rho}{1 - \rho_+}\right) \,, \tag{12}$$

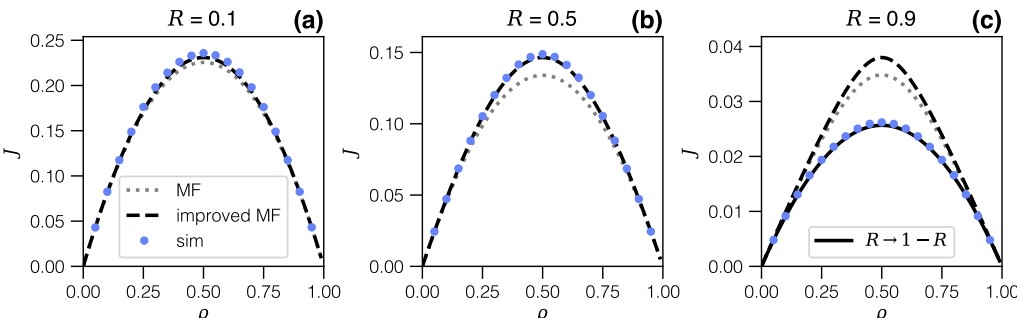

Figure 6: Plots of the current $J = \gamma_- \rho_+$ as in Fig. 3, displaying the three regimes separately. The dotted gray curve represents the mean-field (MF) solution with $\rho_+$ as found in Eq.(8), the black dashed curve is the solution of the improved mean-field from Eq.(13) and the full black curve in panel (c) is Eq. (13) after exchanging the rates $\gamma_-$ and $\gamma_+$ (and thus replacing $R$ by $1-R$), as emphasised in Eq.(14).

where we have used $\rho_- = \rho - \rho_+$ as well as Eq. (11). We again obtain a quadratic equation for $\rho_+$. Picking the physically relevant branch yields a result for $\rho_+$, and hence for the current:

$$\rho_+ = \frac{1}{2}\left(1 - \sqrt{1 - 4R\rho(1-\rho)}\right) . \tag{13}$$

This solution, which only depends on the density $\rho$ and on the ratio of rates via $R$, is superposed in Fig. 6 as a black dashed line. It is in excellent agreement with simulation data for values up to $R \simeq 1/2$ (panel 6.b). However, it still fails for yet higher expansion rates (panel 6.c).

Before moving on it seems interesting to mention that the method of reducing the lattice by eliminating those sites which do not participate in the dynamics can also be applied to recover the solutions of the model known as $\ell$-TASEP with extended particles, as presented in [1, 22]. If in fact we consider particles of size $\ell$ with density $\rho$ advancing one site at a time at a rate $\gamma$, then $(\ell-1)$ sites are not relevant to the dynamics, and for this case the appropriate reduced density reads $\tilde{\rho} = N/[L-(\ell-1)N] = \rho/[1-(\ell-1)\rho]$. Eq. (11) can then be written as $J = \gamma\rho(1-\tilde{\rho}) = \gamma\ell\rho(1-\ell\rho)/[\ell\rho(1-\ell)+\ell]$, as in [1, 22]. This mapping thus provides a straightforward mean-field method which allows us to avoid more complicated combinatorial arguments. Below we will exploit this method further.

## 3.5 Symmetry argument for $R \geq 1/2$

In the regime dominated by particle expansion the microscopic picture becomes increasingly complex: both compressed and expanded particles are expected to be present in significant proportions, as expansion is favoured but will often fail due to lack of available space. We are thus dealing with a complex crowded system, mixing compressed and expanded particles, and excluded volume interactions are expected to entail correlations. Indeed, given that in this scenario the rate-limiting step is particle contraction, the typical locally blocking configuration consists of an expanded particle followed by a contracted particle: as long as the former does not contract, the latter cannot expand. A similar situation is found with particles having an internal state but no conformational change [18], where active particles form clusters behind blocking inactive ones. Accounting for such significant correlations typically requires a much more involved analysis, but fortunately this is not necessary.

Instead, we appeal to another symmetry, noting that the current remains unchanged under the transformation $\gamma_\pm \longleftrightarrow \gamma_\mp$. To see this, we examine again Fig. 5.b. Indeed, reading the time-reversal of the process, i.e. from bottom to top (upwards arrows at the right-hand side), shows that the reverse current of holes (|) can also be described as a contraction-expansion process in

exactly the same fashion as the current of particles. *However*, the expansion and compression steps now arise in reverse order, which amounts to exchanging the rates $\gamma_{\pm} \leftrightarrow \gamma_{\mp}$, which is equivalent to the change $R \leftrightarrow 1-R$.

Since the expanded conformation (+) in the evolution is shared between particles (−) and holes (|), we can thus use Eq. (13), provided we substitute $R$ by $1-R$. From the previous section we know that the improved mean-field expression successfully predicts $\rho_+$, and therefore the current in the regime $R \leq 1/2$; upon exchanging $\gamma_{\pm} \leftrightarrow \gamma_{\mp}$, the theory will hence capture the current of holes in the $1-R \leq 1/2$ regime. Given that the current of holes and particles are equal in the steady state, the particle current in the different regimes can therefore be summarised as

$$
J(R; \rho) =
\begin{cases}
\gamma_- \rho_+(R; \rho) & \text{for } R \leq 1/2 \\
\gamma_+ \rho_+(1-R; \rho) & \text{for } R > 1/2 \, .
\end{cases}
\tag{14}
$$

This result is superposed in Fig. 6.c, and it shows excellent agreement with simulation data.

### 3.6 Limiting cases

Contact with other models can be made asymptotically via several limiting cases of Eqs. (13) and (14). A first point is made by Taylor-expanding the square-root term in Eq. (13) for $\rho \ll 1$. Assuming $\rho \ll 1$, developping to linear order yields

$$
J = \gamma_- \rho_+ \approx \gamma_{\text{eff}} \rho (1-\rho) \, ,
\tag{15}
$$

with

$$
\frac{1}{\gamma_{\text{eff}}} := \frac{1}{\gamma_+} + \frac{1}{\gamma_-} \, ,
\tag{16}
$$

which shows that the slope of the current-density relation $J(\rho)$ for small densities is set by an effective rate accounting for the entire cycle, defined as expected for any two-step process as long as crowding plays no role.

At further thought, the Taylor expansion is also justified in two more cases. First, for high densities $1-\rho \ll 1$ is a small parameter, and the same expression thus also fixes the high-density slope to the same (negative) value, reproducing the particle-hole symmetry $\rho \leftrightarrow 1-\rho$. Second, the same expansion is also justified for $R \ll 1$. Therefore Eq. (15) shows that in the regime of very slow particle expansion the asymptotic dynamics is that of an effective TASEP, with an effective rate corresponding to a 2-step process, given by Eq. (16). This rate always applies to isolated particles ($\rho \ll 1$), but here ($R \ll 1$) it sets the dynamics throughout the entire density range.

## 4 General case $(\ell_-, \ell_+)$

We have shown that our analysis can quantitatively capture the behaviour of the system with $\ell_- = 1$ and $\ell_+ = 2$, and we have discussed the symmetries exploited to obtain and rationalise our results. In this section we generalise this approach to arbitrary values of contracted and expanded particle sizes $\ell_-$ and $\ell_+ = \ell_- + \Delta\ell$.

To do so we generalise the procedure introduced above, which consists in formulating the mean-field dynamics based on a mapping to a system from which all those sites which do not impact the dynamics have been eliminated. This amounts to the (instantaneous) mapping where each particle is reduced to a single lattice site. The mapping requires reducing each particle to the size of a single site, i.e. we must remove $(\ell_- - 1)$ sites for the compressed (-)

particles and $(\ell_+-1)$ sites for the expanded $(+)$ particles. This corresponds to a density on the reduced system which is

$$\tilde{\tilde{\rho}} = \frac{\rho}{1-(\ell_+-1)\rho_+ -(\ell_--1)\rho_-} \,, \tag{17}$$

which can be seen as a generalisation of Eq. (10). A more detailed explanation is given in Appendix C.

From this mapping we can establish the current as

$$J = 2J_+ = \gamma_+ \Delta\ell \, \rho_- (1-\tilde{\tilde{\rho}})^{\Delta\ell} \,, \tag{18}$$

where the exponent $\Delta\ell$ accounts for the fact that $\Delta\ell$ free sites are required ahead of a particle to permit expansion. Equating currents as we have done for deriving Eq. (12) this now yields

$$\gamma_-\Delta\ell \, \rho_+ = J = \gamma_+\Delta\ell \, (\rho-\rho_+) \left(1-\frac{\rho}{1-(\ell_+-1)\rho_+ -(\ell_--1)(\rho-\rho_+)}\right)^{\Delta\ell} \,, \tag{19}$$

which is an implicit equation for the partial density $\rho_+$.

For the special case of expansion by a single site ($\Delta\ell = 1$, even with arbitrary $\ell_-$), this equation reduces to a quadratic equation which can readily be solved. For general $\Delta\ell$, Eq. (19) is no longer tractable analytically, but it can be treated numerically to solve for $\rho_+$. The result is shown in Fig. 7, for a selection of choices for $(\ell_-,\ell_+)$. On each plot data points from simulation are confronted to predictions based on solving Eq. (19), superposing three choices of expansion/contraction rates which cover the three regimes ($R = 0.1$, $R = 0.5$ and $R = 0.9$, respectively: note that the symmetry $R \leftrightarrow 1-R$ has been used for the large values of $R$). Agreement with simulation data is excellent. The only significant deviations arise for the high density regions in the case where the expansion length $\Delta\ell$ is large, a scenario in which statistics on the current must be expected to be poor.

Further insight may be gained from making contact with other established models. In the low density regime, the initial slope of the current-density relation is compatible with the model for fixed size particles (of size $\ell = \ell_-$) as well as the two-state model (in the limit of fast activation rates). This follows directly from the fact that all these models share an effective TASEP model as limiting behaviour. Indeed, the low-density regime is that of the effective TASEP, Eq. (15), as expected (and shown explicitly in Appendix B), and therefore equivalent to that of extended particles described in [1,22]. Differences arise as soon as collisions occur.

An asymptotic expansion can also be made as the system approaches full packing. In this case, Eq. (19) can be expanded for large densities, considering both $\epsilon := 1 - \ell_-\rho$ and $\rho_+$ to be small. This yields, based on a power-law ansatz (details are given in Appendix B), an asymptotic relation for the current of

$$J \simeq \gamma_+ \Delta\ell \, \rho^{\Delta\ell+1} (1-\ell_-\rho)^{\Delta\ell} \qquad (\Delta\ell > 1) \tag{20}$$

as full packing is approached ($\rho \to 1/\ell_-$). The direct interpretation is that even though expanded $(+)$ particles are in principle present for any value of $R > 0$, essentially all particles will find themselves in the compressed $(-)$ state, due to crowding. Therefore the physics is asymptotically that of a system of particles of size $\ell_-$ for which the contraction rate no longer plays a role. The fact that particle expansions, and therefore displacements, become exponentially unlikely due to steric hindrance causes the horizontal slope (see Fig. 7) for any $\Delta\ell > 1$.

The case of single-site expansion $\Delta\ell = 1$ is set apart, as full packing is approached with a linear slope. In this case the resulting asymptotic behaviour differs:

$$J \simeq \gamma_{\mathrm{eff}}(1-\ell_-\rho) \,, \tag{21}$$

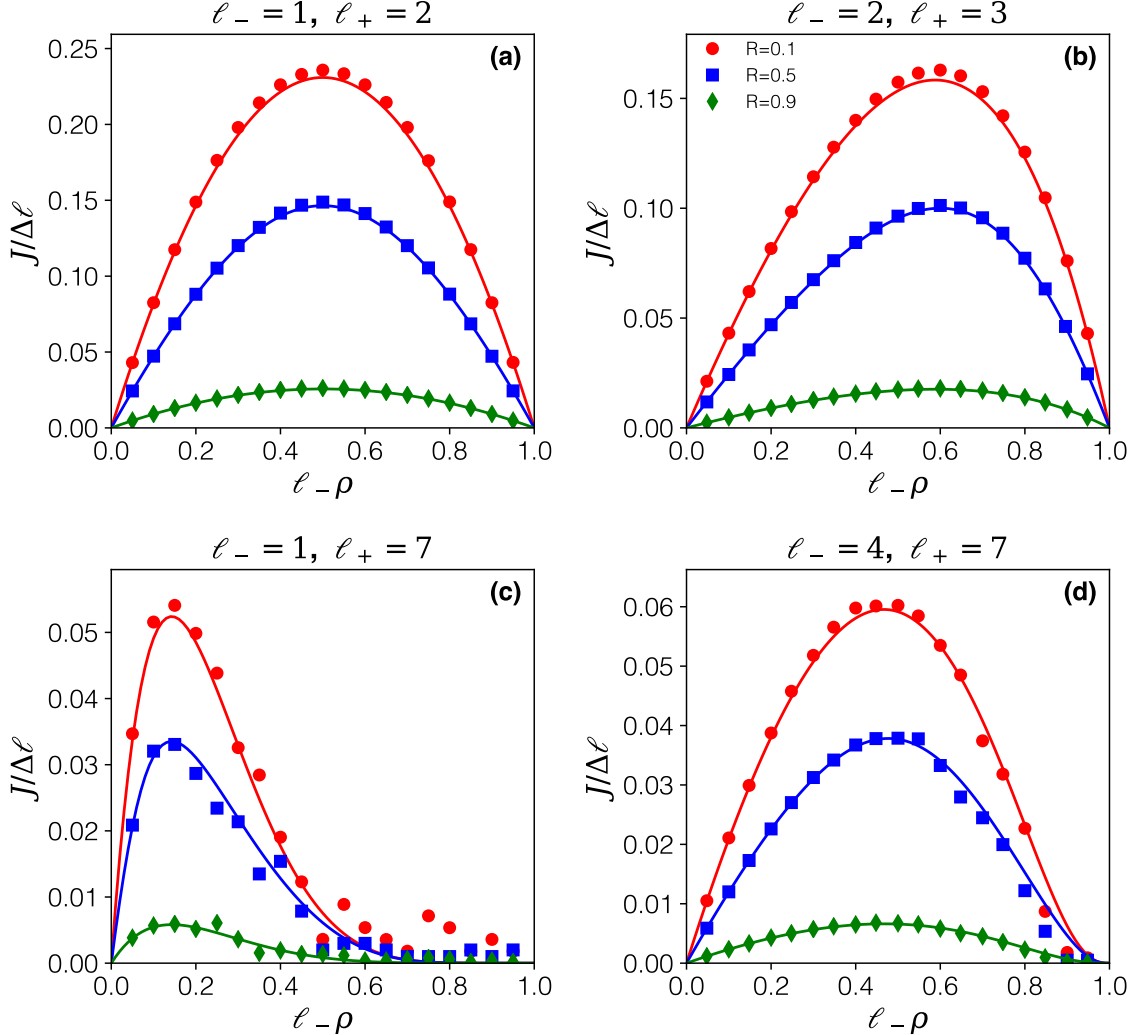

Figure 7: Current-density relation for various choices of $(\ell_-, \ell_+)$, each one contrasting various regimes for the expansion/compression rates ($R = 0.1, 0.5, 0.9$). Data points are from simulations, continuous lines are predictions based on numerically solving Eq. (19). In all plots $\gamma_+ = 1$.

and thus in this case it is the rate of the two-step process, rather than just the expansion rate, which matters. Using this rate the slope is furthermore identical to that found from the corresponding expansion for a $\ell$-TASEP according to MacDonald and Shaw [1,22]. Therefore, if the compression/expansion cycles concern a single site ($\Delta \ell = 1$) the model shares the asymptotic behaviour for large densities with that of fixed-size particles of size $\ell = \ell_-$ and an effective stepping rate given by Eq. (16).

# 5 Illustration via cases of biological interest

In this last section we discuss the general solution for a few specific parameter sets $(\ell_-, \ell_+)$ selected on biological grounds, and we emphasise the differences which the expansion-contraction cycle causes with respect to standard TASEP with extended particles. The biological systems of interest are motor proteins and ribosomes.

The first case we analyse concerns motor proteins advancing on their uni-dimensional substrate, with two scenarios for the stepping mechanisms, as depicted in Fig. 1: some motors advance in a hand-over-hand fashion (e.g. kinesin-1 and myosin V [21]), whereas others follow an inchworm stepping cycle [19, 21]. Within our model these map onto the choices $\ell_- = 1, \ell_+ = 2$ (hand-over-hand) and $\ell_- = 2, \ell_+ = 3$ (inchworm), respectively, assuming that motor heads occupy one site of a discrete lattice that also corresponds to the step-length of the motor.

To assess the impact of the expansion-compression cycle we contrast predictions from the model presented here to those for fixed-size particles. Specifically, we now revert to examining actual currents (rather than the partial density $\rho_+$) as a function of the actual particle density (rather than the renormalised density $\ell_- \rho$), as these are the quantities which can be measured in experiment [5]. For the sake of simplicity we focus on the case $R = 0.5$, which amounts to assuming that both heads of the motor behave equivalently in the stepping cycle. Fig. 8.a compares the solution for hand-over-hand motion ($\ell_- = 1$, $\ell_+ = 2$) to that for inchworm motion ($\ell_- = 2$, $\ell_+ = 3$). We also superpose predictions for the regular TASEP with fixed size particles ($\ell$-TASEP with $\ell = 1$ and $\ell = 2$), using the effective stepping rate Eq. (16), which is the effective rate for the two-step process in the absence of crowding. As is clear from Fig. 8.a, the stepping dynamics with expansion/compression cycles produce fundamental diagrams $J(\rho)$ which differ significantly according to the stepping mechanism used. When a simplified description in terms of constant-size particles ($\ell$-TASEP) is attempted this leads to moderate deviations at intermediate densities, but produces the correct asymptotic behaviour for both small and large densities. More interesting though, we underline that it is standard TASEP ($\ell = 1$) which is generally used to model collective motion of motor proteins. This leads to current-density relations which are in semi-quantitative agreement with hand-over-hand dynamics. However, they differ strongly from results for the inchworm stepping mechanism. Acknowledging the contraction-expansion cycle might therefore prove crucial at least for this latter case.

The second case of biological interest is mRNA translation, where TASEP-based models are often used to describe the collective movement of ribosomes on mRNA strands. Here too, the ribosomes undergo specific structural rearrangements during elongation [20], along with their footprint, which respectively covers around $20 - 22$ and $28 - 30$ nucleotides. As the lattice is based on codons, corresponding to 3 nucleotides, we map these footprints to $\ell_- = 7$ and $\ell_+ = 10$ lattice sites, respectively. In Fig. 8.b we compare the resulting current-density relations of our model to those from 3-TASEP for extended particles with $\ell = 10$ and hopping rate according to Eq. (16), which is generally used to model mRNA translation. Qualitative and quantitative changes between the two models arise. The presence of a compressed state authorises larger densities when compared to fixed-size particles, which implies significant differences in the current-density relation. However, even for intermediate densities there is an impact, which shifts the current maximum to larger densities while reducing the maximum flow at the same time. It is furthermore apparent that the impact at moderate densities is higher for small values of $R$. Comparing the predictions for the case $(\ell_-, \ell_+) = (7, 10)$ for different values of $R$ reveals another interesting point: although the ribosome footprint remains unchanged, the current (and hence the expression rate) which can be achieved is strongly dependent on the dynamics of the stepping cycles.

# 6 Discussions

We have addressed the issue of collective effects in active stochastic transport of objects which undergo cyclic conformational changes during the stepping process. Specifically, we have considered cycles of two conformations occupying different footprints on the supporting track,

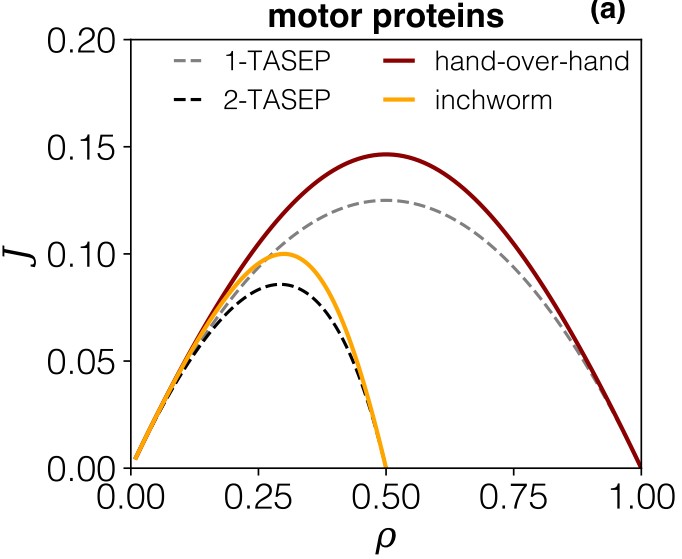

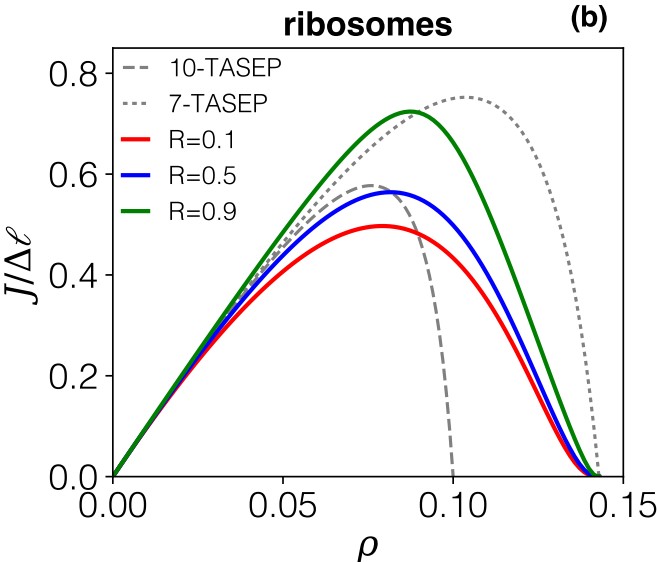

Figure 8: Current $J$ as a function of particle density $\rho$. Standard TASEP with extended particles of size $\ell = 1$ and $\ell = 2$ are plotted with dashed lines (gray and black line respectively) in panel (a). In panel (a) we also plot the two mechanisms (hand-over-hand: $\ell_- = 1, \ell_+ = 2$, and inchworm: $\ell_- = 2, \ell_+ = 3$) for $R = 0.5$. In panel (b) we show the current for the inchworm-like movement of the ribosomes $\ell_- = 7, \ell_+ = 10$, for different values of $R$ and setting the time scale by fixing the effective rate to $\gamma_{\text{eff}} = 10/s$, which is the usually accepted rate for ribosome stepping.

represented here through particle sizes specific to each conformation. These questions are directly motivated by the stepping mechanisms of motor proteins moving along actin filaments or microtubules, as well as ribosomes moving along mRNA strands.

The current that can be sustained by such particles at a given density is strongly affected by the footprint of each conformation on the substrate. This is a consequence of the fact that the expansion step is subject to crowding effects, as it requires sufficient space ahead of a particle to be able to expand. At the same time, the rates for particle expansion/contraction affect the average particle size, and thus retro-effect crowding. We have analysed the full process, coupling these mutual dependencies, on a periodic lattice. We have deduced analytical expressions for the current-density relation $J(\rho)$. Various regimes arise.

If particle expansion is the rate-limiting step, straightforward mean-field arguments work well, just as is the case for the standard TASEP: indeed, an effective TASEP model is asymptotically recovered in the limiting case where $\gamma_+ \ll \gamma_-$.

For intermediate regimes, where $\gamma_- \simeq \gamma_+$, a more refined argument is required. For this we have established a mapping to a reduced system, based on which results from standard TASEP can be adapted. This provides excellent results for the entire density range when compared to data from numerical simulations. Our approach, which can also reproduce previous results of the TASEP with extended particles [1, 22], does not require a combinatorial analysis.

Finally, the regime of fast expansion ($\gamma_+ \gg \gamma_-$) is microscopically very different from the previous regimes whenever the particle density is not small: the tendency of particles to remain in their expanded state increases crowding, which ultimately self-limits the possibility for expanding. Despite the bias towards expansion, here we are thus necessarily dealing with a complex mixture of expanded and compressed particles, which makes a microscopic description difficult. However, the problem is implicitly solved by exploiting a symmetry with respect to inverting the expansion/compression rates, which has allowed us to adapt the expression for slow expansion to also cover this regime. Predictions are again in excellent agreement with numerical simulations.

Hand-over-hand motion between consecutive sites ($l_- = 1, l_+ = 2$) is special in that it gives rise to an additional symmetry: for this case, and this case only, transport of advancing particles can equivalently be interpreted in terms of receding holes, which follow identical dynamical rules. Therefore the current-density relation reflects this symmetry, $J(\rho) = J(1-\rho)$. The fact that this result is preserved from standard TASEP is surprising: our model combines features from models involving an internal state in particle dynamics [16, 17] and particles larger than a single lattice site [1, 22]; each of these changes separately leads to a skew $J(\rho)$ relation with a maximum shifted to densities above 1/2, yet combining them re-establishes the symmetry, as we have argued microscopically.

We were then able to generalise our approach to conformations occupying an arbitrary number of sites $\ell_-$ and $\ell_+$, and our solution successfully reproduces the outcome of simulations.

We have explored the implications of our model by comparing the fundamental diagrams $J(\rho)$ of different conformational stepping cycles to those from the corresponding effective $\ell$-TASEP with constant size particles, confronting two classes of biological motors. This shows that there is a quantitative difference when explicitly considering the particles' conformational changes. The implication for motor proteins is that the standard TASEP model which is commonly used may not be well suited for modelling inchworm motion, for which the varying footprint along the conformational cycle significantly modifies the current-density relation. For ribosomes, where there is no reason to assume equal rates for expansion and contraction steps, we have shown that this asymmetry between rates affects both the optimal translation rate and the density at which it is achieved. Future work should clarify finer points, such as for example the fact that, despite advancing a single codon (site) after a full cycle, ribosomes may

transit through intermediate configurations which have a larger footprint [20], which would require extending the model introduced above.

# A  Numerical Simulations

The dynamics of the system is simulated following the Gillespie scheme [23]. For each particle density $\rho$, and for given values of the rates $\gamma_\pm$, a timescale $\tau$ is fixed as $\tau := \max\{\gamma_+, \gamma_-\}$. Particles then move according to the dynamics explained in Section 2. We set a transient time $T = 100 \times \rho L \tau$, during which we do not collect data, and then compute averages over the time interval $T < t < 3T$.

For each simulation point in the figures an initial condition has been generated, corresponding to the desired density $\rho$, and hence particle number $N$, according to the following protocol:

(i) create a tentative initial condition: iteratively place a particle in the compressed (-) state on sites 1,2,... with probability $\rho$, respecting volume exclusion (periodic boundary conditions are applied if necessary), until one of the following conditions is met: either the lattice is populated with $N$ particles or a total number of $L \times 10^3$ insertion attempts have been made.

(ii) accept or reject the configuration: reject this tentative initial condition if the total density which has been achieved is beyond a tolerance of the targeted density, or if it represents a frozen state (no particles can move) then go back to point (i).

(iii) start the simulation to relax to the stationary state, as described above.

Statistics could be improved by averaging over initial conditions, but this has not been necessary.

# B  Asymptotics

This appendix expands on deriving the asymptotic behaviour in various limiting cases. The starting point is the improved mean-field relation Eq. (19), which we recall here for convenience, written in a slightly more convenient fashion:

$$\rho_+ = \frac{\gamma_+}{\gamma_-}(\rho - \rho_+)\left[\frac{(1-\ell_-\rho)-\Delta\ell\,\rho_+}{\rho+(1-\ell_-\rho)-\Delta\ell\,\rho_+}\right]^{\Delta\ell}. \tag{22}$$

**Special case** $(\ell_-, \ell_+) = (1,2)$  For this simplest case the solution Eq. (13) can be Taylor expanded if the second term under the square root is small, which is the case at the edges of the density interval ($\rho \ll 1$ or $1-\rho \ll 1$), as well as in the case of slow expansion ($R \ll 1$). Using either as a small parameter and truncating after the linear term yields

$$\rho_+ = \frac{1}{2}\left(1 - (1 - \tfrac{1}{2}4R\rho(1-\rho) + ...)\right) \simeq R\rho(1-\rho), \tag{23}$$

from which follows the asymptotic current

$$J \simeq \gamma_-\rho_+ \simeq \frac{\gamma_-\gamma_+}{\gamma_-+\gamma_+}\rho(1-\rho). \tag{24}$$

In particular, the low-density current is $J \simeq \gamma_{\text{eff}} \rho$, with an expected rate given by Eq. (16), as expected for a two-step process for which the expansion/compression dynamics is governed by the corresponding rates rather than by crowding effects. The result for the current also covers the high-density regime, as the expansion also holds for $1 - \rho \ll 1$, or simply from the particle-hole symmetry.

**General case** $(\ell_-, \ell_+)$   The low-density expansion goes through in the general case, considering both $\rho$ and $\rho_+$ as small parameters. Eq. (22) can be expanded as

$$\rho_+ = \frac{\gamma_+}{\gamma_-} (\rho - \rho_+) \left[ \frac{1 - (\ell_- \rho + \Delta\ell \, \rho_+)}{1 - (\ell_- \rho + \Delta\ell \, \rho_+ - \rho)} \right]^{\Delta\ell}$$

$$\simeq \frac{\gamma_+}{\gamma_-} (\rho - \rho_+) \left[ 1 - \rho + \ldots \right]^{\Delta\ell}$$

$$\simeq \frac{\gamma_+}{\gamma_-} (\rho - \rho_+) \left[ 1 - \rho \, \Delta l + \ldots \right],$$

where the first step is to take the denominator be of the form $1/(1 + \ldots)$. To first order we thus have

$$\rho_+ \simeq \frac{\gamma_+}{\gamma_+ + \gamma_-} \rho \, (1 - \rho \, \Delta\ell),$$

and thus the asymptotic current for small densities is again compatible with the effective rate Eq. (16):

$$J \simeq \Delta\ell \, \gamma_{\text{eff}} \rho \, (1 - \Delta\ell \rho). \tag{25}$$

The approach to full packing can be examined by considering the small parameter $\epsilon := 1 - \ell_- \rho$. We note that $\rho_+$ also vanishes as $\epsilon \to 0$, and we proceed by making the ansatz

$$\rho_+ = \alpha \, \epsilon^\nu, \tag{26}$$

with some constant $\alpha$. Then Eq. (22) can be re-expressed, by substituting for both $\rho$ and $\rho_+$, as

$$\rho_+ \left[ \frac{1 - \epsilon}{\ell_-} + \epsilon - \Delta\ell \, \alpha \, \epsilon^\nu \right]^{\Delta\ell} = \frac{\gamma_+}{\gamma_-} \left( \frac{1 - \epsilon}{\ell_-} - \alpha \, \epsilon^\nu \right) \left[ \epsilon - \Delta\ell \, \alpha \, \epsilon^\nu \right]^{\Delta\ell}, \tag{27}$$

to find

$$\rho_+ \left( \frac{1}{\ell_-} + \ldots \right)^{\Delta\ell} = \frac{\gamma_+}{\gamma_-} \frac{1 - \epsilon}{\ell_-} \epsilon^{\Delta\ell} (1 + \ldots).$$

We now equate the leading order terms on both sides, assuming for the moment that $\nu > 1$, to find

$$\nu = \Delta\ell \qquad \text{and} \qquad \alpha = \frac{\gamma_+}{\gamma_-} (\ell_-)^{\Delta\ell - 1}.$$

This implies an asymptotic current of

$$J \simeq \gamma_+ \Delta\ell \, (\ell_-)^{\Delta\ell - 1} \left( 1 - \ell_- \rho \right)^{\Delta\ell} \qquad (\Delta\ell > 1), \tag{28}$$

whenever $\nu > 1$, and thus $\Delta\ell > 1$.

Finally, we observe that the special case $\nu = 1$ changes the asymptotic behaviour in Eq. (27), as all terms in the last parenthesis are now of the same order in $\epsilon$. Equating leading order terms on both sides now leads to the condition

$$\frac{\alpha}{(\ell_-)^{\Delta\ell}} \epsilon = \frac{\gamma_+/\gamma_-}{\ell_-} (1 - \Delta\ell \, \alpha) \epsilon^{\Delta\ell}.$$

This requires $\Delta\ell = 1$, and the resulting equation on $\alpha$ yields

$$\alpha = \frac{\gamma_+/\gamma_-}{1 + \gamma_+/\gamma_-} = \frac{\gamma_+}{\gamma_+ + \gamma_-} \, .$$

Therefore the asymptotic current in the case of single-site expansion steps $\Delta\ell = 1$ is different from the general case,

$$J \simeq \gamma_{\text{eff}}(1 - \ell_-\rho) \qquad (\Delta\ell = 1) \, . \tag{29}$$

In contrast to the case $\Delta\ell > 1$, the relevant rate is here the effective rate given by Eq. (16), which characterises a two-step process. Eq. (29) coincides with the corresponding expansion of the $\ell$-TASEP prediction [1, 22] with $\ell = \ell_-$, i.e. for large densities the asymptotic behaviour with an expansion/contraction cycle is identical to that of fixed-size particles of size $\ell_-$, if the hopping rate is matched to that of a two-step process.

## C  Detailed justification of the general current-density relation

In the main text we have established that the effective density to be used in the general case $(0_-, \ell_+ = \ell_- + \Delta\ell)$ arises from 'eliminating all sites which do not participate in the dynamics'. The purpose of this Appendix is to provide a more detailed justification for Eq. (17), extending the result used in section 3 via a two-step argument.

First, we remark that a mapping can be made to a simplified system of density $\rho'$ in which all contracted particles are reduced to size $\ell'_- = 1$, and thus $\ell'_+ = 1 + \Delta\ell$:

$$J(\ell_-, \ell_+ = \ell_- + \Delta\ell; \rho) = J(\ell'_- = 1, \ell'_+ = 1 + \Delta\ell; \rho'), \tag{30}$$

with

$$\rho' = \frac{N}{L - (\ell_- - 1)N_-} = \frac{\rho}{1 - (\ell_- - 1)\rho_-} \, . \tag{31}$$

This directly reflects the fact that each leading site occupied by a particle is necessarily followed by a further $\ell_- - 1$ occupied sites: these have no impact on any of the compression or expansion processes, as they always occupy the same space, and they can therefore be accounted for through the mapping onto a simplified (primed) lattice where all particles have been reduced do size $\ell'_- = 1$, reducing the overall lattice length by the corresponding number of $N \times (\ell_- - 1)$ sites.

In a second step we can apply a mapping similar to the one used for establishing Eq. (10), and account for the fact that a reduced density must be considered for determining the acceptance probability of an expansion step. Generalising the argument leading to Eq. (10) this is done by eliminating those $N_+ \times (\ell_+ - 1)$ sites covered by the expanded particles which have no impact on the dynamics. Note, however, that the reduced density $\tilde{\rho}$ has to be obtained starting from the modified densities $\rho'$ introduced in Eq. (31) and the correspondingly reduced partial densities

$$\rho'_\pm = N_\pm / (L - N_-(\ell_- - 1)) \, , \tag{32}$$

which amounts to

$$\tilde{\tilde{\rho}} := \tilde{\rho}(\rho', \rho'_-) \, . \tag{33}$$

The effective density after the two successive mappings by Eqs (31) and (33) then yield

$$\rho \to \rho' \to \tilde{\tilde{\rho}} = \tilde{\rho}(\rho'; \rho'_-) \, , \tag{34}$$

which can be cast into the form of Eq. (17).

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
