# Peer review of "Exclusion process subject to conformational size changes"

_SciPost Physics, doi:SciPost Phys. 6, 077 (2019)_

## Round 1 · Referee Report · Anonymous · 2019-6-7

Report
The article presents an exclusion process on a one-dimensional grid motivated by biological systems. It is like a totally asymmetric exclusion process, but the particles have two modes of advancing: stretching and contracting like a worm or doing "somersaults" - which is like contracting and then stretching. The phenomenology depends quite strongly on the ratio of lengths. For the particular case of ratio lengths 2-1 the authors uncover a nontrivial particle-hole symmetry, otherwise this symmetry is lost. The authors develop a modified mean-field approximation that gives very good results. Finally, they discuss the consequences in the biological context. The paper is very well argued and convincing. I recommend its publication.
Anonymous on 2019-06-11 [id 538]
This is an interesting article, with numerical experiment that give light on the behaviour of the transport associated to various stationary regimes: high and low density, slow and fast expansion....
The main question, that actually is not explicitely stated, is about the structure of the stationary states for this dynamics. In the usual TASEP these are given by Bernoulli measures, which is the reason why the mean-field prediction is 'exact' there. In the present model the fact that the mean field prediction is not accurate for fast expansion rate implies that Bernoulli measure are not the stationary measure, but a good approximation for slow expansion regime. I do not find the argument of the improved mean field very clear, and the fact that the associated prediction for R=0.9 is worse than the mean field one is suspicious.
In conclusion I think this article will be useful for a further mathematical analysis of the dynamics, and it deserves to be published.

---

## Editorial Decision

published